# Fluorometric Quantification of Human Platelet Polyphosphate Using 4′,6-Diamidine-2-phenylindole Dihydrochloride: Applications in the Japanese Population

**DOI:** 10.3390/ijms22147257

**Published:** 2021-07-06

**Authors:** Taisuke Watanabe, Yutaka Kitamura, Hachidai Aizawa, Hideo Masuki, Tetsuhiro Tsujino, Atsushi Sato, Hideo Kawabata, Kazushige Isobe, Koh Nakata, Tomoyuki Kawase

**Affiliations:** 1Tokyo Plastic Dental Society, Kita-ku, Tokyo 114-0002, Japan; watatai@mui.biglobe.ne.jp (T.W.); shinshu-osic@mbn.nifty.com (Y.K.); sarusaru@mx6.mesh.ne.jp (H.A.); hideomasuki@elm-dc.com (H.M.); tetsudds@gmail.com (T.T.); atu.net@nifty.com (A.S.); hidei@eos.ocn.ne.jp (H.K.); kaz-iso@tc4.so-net.ne.jp (K.I.); 2Bioscience Medical Research Center, Niigata University Medical and Dental Hospital, Niigata 951-8520, Japan; gentle@med.niigata-u.ac.jp; 3Division of Oral Bioengineering, Niigata University Graduate School of Medical and Dental Sciences, Niigata 951-8514, Japan

**Keywords:** polyphosphate, DAPI, platelet, fluorometry, gender difference

## Abstract

Polyphosphate (polyP), a biopolymer of inorganic phosphate, is widely distributed in living organisms. In platelets, polyP is released upon activation and plays important roles in coagulation and tissue regeneration. However, the lack of a specific quantification method has delayed the in-depth study of polyP. The fluorescent dye 4′,6-diamidine-2-phenylindole dihydrochloride (DAPI) has recently received attention as a promising probe for the visualization and quantification of cellular polyP levels. In this study, we further optimized quantification conditions and applied this protocol in quantification of platelet polyP levels in a Japanese population. Blood samples were collected from non-smoking, healthy Japanese subjects (23 males, 23 females). Washed platelets were fixed and probed with DAPI for fluorometric determination. PolyP levels per platelet count were significantly higher in women than that in men. A moderate negative correlation between age and polyP levels was found in women. Responsiveness to CaCl_2_ stimulation was also significantly higher in women than that in men. Overall, our optimized protocol requires neither purification nor degradation steps, reducing both the time and bias for reproducible quantification. Thus, we suggest that despite its low specificity, this DAPI-based protocol would be useful in routine laboratory testing to quantify platelet polyP levels efficiently and economically.

## 1. Introduction

Polyphosphate (polyP), a linear, unbranched polymer of orthophosphate residues linked by phosphoanhydride bonds, is ubiquitously found in nature. Its chain length can vary from a few phosphate units to several thousand depending on the organism [1]. The synthesis of polyP depends on the metabolic status of the cell, and polyP levels are correlated with cellular ATP levels [2]. Thus, tissues with a high energy demand and a high regeneration or proliferation capacity are rich in polyP [3,4]. As for the pathophysiological functions of platelets, polyP, which is stored in dense granules [5,6], stimulates an array of procoagulant mechanisms and drives fibrin formation through the activation of factor XII (FXII) and factor XI (FXI) and the conversion of factor V to its active form [7]. Furthermore, it has often been reported that polyP acts as a proinflammatory factor to stimulate bradykinin formation and increase vascular permeability [8,9]. As a secondary effect, polyP in the plasma is thought to be degraded by tissue-nonspecific alkaline phosphatase, which is expressed in liver, kidney, and skeletal tissue, to control the concentration of orthophosphate available for apatite mineral formation [10]. Therefore, polyP is thought to be involved in several major physiological functions. However, due to the lack of standardized quantification methods, both platelet polyP levels and dynamic mobilization have rarely been reported and cannot be compared using data obtained by different quantification methods [11].

To date, many techniques have been developed for the detection of polyP [12]. However, the main techniques are staining techniques, followed by light and fluorescence microscopic examination. Among the dyes used for polyP staining, the most widespread is 4′,6-diamidine-2-phenylindole dihydrochloride (DAPI). The best advantage of DAPI is that it does not require prior purification nor other processing such as brief boiling or digestion with proteinase K [13]. Moreover, the ability of DAPI to bind to a large variety of molecules may hamper the detection of low amounts of polyP in mammalian cells. Thus, low specificity could be a disadvantage. However, fluorescence emission from free DAPI to DAPI-DNA is minimal at 550 nm. Thus, the DAPI-polyP signal could be detected as highly specific and essentially independent of the presence of DNA [14]. Although DAPI also binds to lipids, showing a similar yellow fluorescence, DAPI-lipids can be distinguished using a relatively longer incubation time [12]. 

To overcome this disadvantage, more specific fluorescent dyes, namely JC-D7 and JC-D8, have been developed. The notable advantage of these dyes is the lack of nonspecific binding to Pi-containing molecules, such as nucleic acids, nucleosides, and sodium phosphate [15]. However, their affinity to polyP is lower than that of DAPI [15]. In addition, they also bind to heparin and are not always commercially available, that is, they are made-to-order products [16]. Therefore, at present, DAPI is superior to other dyes in terms of commercial availability, sensitivity, and specificity. In a previous cytochemical study, we demonstrated that the optimized DAPI-staining protocol can be used to easily visualize the distribution of polyP in platelets and the extracellular spaces in a semi-quantitative manner [17]. In contrast, due to the need for extraction and purification, techniques for polyP quantification, such as RAMAN microscopy and enzyme assays, have not been accepted as routine laboratory methodologies [12]. Although flow cytometry is a powerful tool for the quantification of polyP-positive platelets, it is not ideal for the quantification of polyP.

We have long investigated the clinical applications of platelet concentrates (PCs) in regenerative therapy and their quality assurance [18,19,20,21,22,23,24,25]. Growth factors and fibrinogen contained in platelets are major factors involved in tissue regeneration. In a previous study [17], however, we confirmed that activated platelets secrete polyP along with the growth factors [26]. If polyP has a potent ability in tissue repair and regeneration [10,27], we expect PCs containing significant amounts of polyP to be applicable for regenerative therapy, as shown previously [28], and hypothesized that polyP may be utilized as a marker of the quality of PCs, i.e., a marker of their predictable clinical outcomes. In the present study, we aimed to optimize the polyP quantification method using DAPI for reproducible data acquisition and quantify the polyP levels in a Japanese population to construct a basic dataset.

## 2. Results

The age distribution of each group is shown in Figure 1. The mean ages of the male and female groups were 43.7 ± 13.1 and 39.1 ± 13.4, respectively (Figure 1a), while the median values were 47.0 and 36.0, respectively (Figure 1b). However, no significant differences were observed in the age distribution between the male and female groups. 

The affinities of DAPI (Dojindo, Kumamoto, Japan) to polyP and ADP are shown in Figure 2a,b. When 4 μg/mL DAPI was used, more intense fluorescence signals were obtained in polyP (832 A.U. at 1.56 μg/mL) than those in ADP. A clear linear regression (coefficient of determination: R^2^ = 0.999) was observed for polyP concentrations up to 1.56 μg/mL. In contrast, the fluorescence signals were weaker in ADP (256 A.U. at 427 μg/mL), and a clear linear regression was observed in a limited range of lower ADP concentrations (up to 26.7 μg/mL).

The effects of CaCl_2_ (included in the assay solution) on polyP quantification are shown in Figure 2c,d. In the given range of platelet polyP levels (0.6–1.0 μg/mL), CaCl_2_ at concentrations up to 10 ppm (0.001%), which may correspond to the theoretical maximum concentration of CaCl_2_ remaining in the 0.1% CaCl_2_-stimulated platelets, somewhat amplified DAPI signals. This amplification was approximately 15–25% higher than that in the controls in the absence of added CaCl_2_.

The factors influencing the quantification of platelet polyP levels are shown in Figure 3. First, the relationship between the platelet count and DAPI-dependent fluorescent signals was examined (Figure 3a). A good linear regression was obtained for up to 5 × 10^7^ platelets, with an R^2^ value of 0.995. When the DAPI concentrations increased, the range of linear regression could be expanded. Next, the effects of preservation days of whole-blood samples on platelet polyP levels were examined (Figure 3b). A significant reduction was observed in the first 24 h, however, thereafter, this reduction was suppressed. Finally, the effects of white blood cell (WBC) inclusion on polyP levels were examined (Figure 3c). Platelet suspensions including WBCs were prepared by incorporating a buffy coat into a pure platelet-rich plasma fraction according to the protocol of leukocyte-rich PRP preparation. Since RBCs were included into the platelet suspensions along with WBCs (hemoglobin (HGB) ≥ 0.1 g/dL) (Figure 3d) and since HGB absorbs the red light of 425 nm, the fluorescence intensity was significantly reduced. At least, the fluorescence intensity was not increased by DNA contained in WBCs. A strong negative correlation between WBC counts and platelet polyP levels (absolute correlation coefficient: |R| = 0.728) was observed.

The effects of donor age on the platelet polyP levels in the male and female groups are shown in Figure 4. The negative correlation was very weak (|R| = 0.0785) and moderate (|R| = 0.411) in the male and female groups, respectively. When both groups were compared, the female group showed significantly higher platelet polyP levels.

The effects of exogenously added CaCl_2_ on platelet polyP levels are shown in Figure 5. Platelet responses were stopped by fixation at 15, 30, and 60 min of incubation. CaCl_2_ reduced the platelet polyP levels in a dose and time-dependent manner. When both groups were compared, platelets prepared from female donors were more sensitive to Ca^2+^ stimuli and reduced polyP levels more quickly than those prepared from male donors. In the case of samples forming fibrin clots, platelet polyP was almost depleted. However, in many cases, approximately 30% of polyP was retained in the platelets. At 60 min of incubation, these quantification data were roughly consistent with visualization data, showing the existence of DAPI-stained polyP particles retained in platelets.

## 3. Discussion

### 3.1. Optimization and Validation of the Quantification Protocol

The main findings of this study were the gender and age differences observed in the platelet polyP levels. However, since there were several difficulties to overcome, it should first be noted how the quantification protocol was optimized to ensure a better reproducibility. The difficulties lie in the specificity of the probe and the nature of polyP. As described earlier, due to the simple repeated structure of polyP and its heterogeneous nature, no specific probes, such as monoclonal antibodies, have been developed, which comprises the first difficulty. Among the probes and modalities developed thus far, DAPI is the most promising and widely used probe for qualitative visualization. Second, polyP chains contained in platelets are relatively shorter than those of bacteria and are easily cleaved by plasma phosphatases. Furthermore, polyP is not tightly fixed onto the plasma membrane or intracellular organelles, even after long fixation with ThromboFix. Through repeated unsuccessful experiences by us and other researchers, we have learnt that the first difficulty cannot be easily solved, and may, in fact, not be solved in the near future.

Thus, we concentrated our efforts on the second matter, and attempted to reduce possible technical biases by simplifying the protocol. As described in the Introduction, protocols requiring extraction and/or purification processes amplify technical errors and are thus not suitable for the routine laboratory testing of samples. Since protocols using DAPI do not require such processes, we chose to optimize this protocol. Nevertheless, several issues remain: Other biases could be easily produced by the operators’ skills. For a reproducible quantification, plasma and buffers containing phosphates should be completely excluded from the samples. However, when platelets are intensively washed by repeated pipetting and centrifugation, platelet polyP, especially membrane-bound polyP, is easily released, resulting in a significant reduction in quantified polyP values. Therefore, we decided to limit the washing process, despite this simplified protocol potentially allowing for inclusion to increase the polyP values. Even though the operators’ technique may influence quantification, we hope that the accumulation of data will reduce technical-sensitive variations and feedback to further refine the protocol for standardization.

DNA, RNA, and other Pi-containing molecules, such as amorphous calcium phosphate and inositol phosphates [29], may influence quantification. To address this, pure-PRP, which does not include white blood cells or other nucleated cells, was prepared and subjected to quantification. Although Pi-containing molecules could not be completely excluded, buffers were replaced with Milli-Q water (Merck, Darmstadt, Germany). Precipitated calcium phosphate was theoretically not present in this assay. Although inositol phosphates are generated in activated platelets in a receptor-operated manner [30], this increase is transient and is rarely utilized for mineralization. Thus, it is possible that platelet inositol phosphates significantly influence polyP quantification. Regarding possible degradation by phosphatases, we did not find that orthovanadate (1 mM) significantly increased polyP values. As a result, the exogenous addition of 0.1% CaCl_2_ to the platelets reduced platelet polyP levels in a concentration-dependent manner, and the amount of polyP remaining in the platelets approximately corresponded to the cytochemical visualization data shown in the present and previous studies [17].

Regarding other possible interfering molecules, heparin and ionized calcium should also be considered [29]. In the preparation of PRP for clinical use, citrate-based anticoagulants, such as sodium citrate and A-formulation of acid-citrate-dextrose (ACD-A), are generally recommended [31]. In this study, we adopted ACD-A, which chelates calcium in plasma [32]. In contrast, since we used CaCl_2_ for the activation of platelets, we directly examined the possible effects of ionized calcium on the standard curve prepared using synthetic polyP. Additional CaCl_2_ (0.625–10 ppm) reproducibly increased DAPI signals by 15–25% compared with the controls. We did not quantify the levels of ionized calcium remaining in the platelet pellets after aspiration of the supernatant. However, this type of potential quantification “artifacts” should be carefully considered to avoid misinterpretation of data in studies involving biological samples.

Therefore, although some possible limitations exist and cannot be overcome yet, we conclude that our simplified protocol using DAPI has been sufficiently validated for application in subsequent clinical studies.

### 3.2. Interpretation of the Main Findings of Gender and Age Differences

In this study, the platelet polyP levels were found to be significantly higher in female donors than those in male donors, and were slightly reduced by aging in female donors.

Compared with bacteria, polyP synthetic and degradation pathways are poorly understood in mammalian cells. Our current understanding can be summarized as follows: The polyP levels in mammalian cells, not limited to platelets, depend on the metabolic state of the mitochondria [2]. With regards to gender differences, the number and volume of mitochondria have been reported to be higher in the muscle cells of female mice than male mice [33,34]. In humans, mitochondrial function in peripheral blood mononuclear cells and the brain has been found to be higher in female than that in male participants [35], which may explain this phenomenon. The elevated 17-β estradiol in females is believed to promote the induction of genes regulating mitochondrial function [36]. 

With regards to aging, it is well known that dysfunctional mitochondria, which generate less ATP, are observed in aged organs. This dysfunction is explained by morphological and biochemical features, including a reduced mitochondrial content, altered mitochondrial morphology, and reduced activity of the complexes of the electron transport chain [37]. Thus, our data are supported by these background data.

Considering the involvement of polyP in coagulation, it is also interesting to briefly discuss gender and age differences in blood clotting. It is well known that exogenously replaced female hormones are a risk factor for thrombosis [38,39,40]. Since excessive blood clotting is also influenced by other risk factors, such as smoking, being overweight, prolonged bed rest, long periods of sitting, and cancer [41], it may be shortsighted to directly connect our findings with thrombosis. However, our findings may partially reflect these clinical backgrounds.

In terms of platelet sensitivity to exogenous CaCl_2_, the female samples showed a quicker reactivity than the male samples did. This finding is consistent with experimental data obtained from animals regarding the sensitivity to agonists [42]. However, there are contradictory reports showing that platelet adhesion and aggregation on collagen or fibrinogen under shear flow is equivalent between female and male mouse platelets [43]. Thus, this will need to be investigated in more detail in future studies.

### 3.3. Relevance to the Quality of Platelet Concentrates

Based on the wound-healing capacity of platelets, PCs that are represented by PRP were first applied in bone regenerative therapy [44]. Since polyP plays key roles in blood coagulation and inflammatory response, polyP was expected to promote tissue regeneration in concert with platelet-stored growth factors. This expectation was soon realized [10,27]. To date, the therapeutic application of polyP has been investigated most often in bone and cartilage regeneration [10,45]. Recently, polyP appeared to also promote epithelial regeneration as an independent factor [28].

These findings imply the possibility that polyP may be considered a marker of PC quality. Compared with growth factor quantification, polyP quantification can be performed using a simple protocol without the need for high-specialty devices or expensive running costs. The only drawbacks are the requirement for long-term fixation and a dependence on the operators’ skills. In addition to the limitation of washing platelets, inclusion of WBCs along with RBCs should be excluded by pipetting. Although the presence of WBCs may not hamper quantification [14,46], the presence of RBCs, that is, red hemoglobin, appears to reduce the polyP values by absorbing the excitation layer (425 nm). 

Some commercially available synthetic polyP is Ca^2+^-free. Therefore, when applied in biological systems, synthetic polyP acts as a chelator of Ca^2+^ and does not promote coagulation or induce cellular responses, as observed in natural polyP [4]. This may, to some extent, delay the study on the regenerative effects of polyP.

The factors thought to be the most potent in tissue regeneration are growth factors, such as TGFβ and PDGF, in platelets. However, a consensus on gender and age-related differences is yet to be achieved. Some studies have reported that aging reduces growth factor levels in PRP [47,48], while others have reported the absence of or negligible effects of age [49,50]. Similarly for gender differences, two studies have provided conflicting results [47,50], while two other studies have shown absent or negligible effects [48,49]. Although it is difficult to identify the cause of these inconsistencies, experimental protocols, technical biases, populations’ individual differences, and/or circadian variations are potential factors hampering the acquisition of reproducible data.

In this context, the data presented in the current study on the platelet polyP levels should be examined by other groups on populations with larger sample sizes using a standardized protocol for a reliable and convincing conclusion to be reached.

## 4. Materials and Methods

### 4.1. Blood Collection

Blood samples were collected from 46 non-smoking, healthy Japanese donors who were not receiving continuous medical treatment at the time (23 males and 23 females). More details of the donors’ constitution are described in the Results section. The sample size calculation is described in the following subsections. 

Blood withdrawal was performed by venipuncture of the median cubital vein using 21G wing needles (NIPRO, Osaka, Japan). Blood was collected into vacutainer A-formulation of acid-citrate-dextrose (ACD-A) glass tubes (BD, Franklin Lakes, NJ, USA). Blood withdrawal was performed randomly during the daytime depending on the donors’ convenience. The blood samples were intermittently rotated at room temperature (18–22 °C) until use.

The study design and consent forms for all procedures (project identification code: 2019–0423) were approved by the Ethics Committee for Human Participants at the Niigata University School of Medicine (Niigata, Japan) and complied with the Helsinki Declaration of 1964, as revised in 2013.

### 4.2. Sample Calculation

The sample size required for gender differences was calculated using G *Power 3.1.9.7 (Heinrich-Heine-University, Düsseldorf, Germany). Based on the results of a pilot study with six samples (three samples for each group), the tentative effect size was calculated to be 0.8571. To achieve sufficient (≥0.8) power (1-Type II error probability (β)), a minimum of 23 samples were required for each group in a two-tailed *t*-test. 

As for linear regression for platelet polyP level against age, Power (1−β) was calculated by post hoc testing using the G *Power software. When the effect sizes were 0.5 and 0.3, the power to detect true effects was calculated as 0.9692 (>0.8) and 0.5503 (<0.8), respectively. Thus, the sample size (*n* = 23) calculated above could be considered insufficient for statistical analysis.

### 4.3. Preparation of Platelet Suspension and Activation

Prior to centrifugation, an excess amount of ACD-A (0.5 mL) (Terumo, Tokyo, Japan) was added to each blood sample and incubated for 10 min to prevent coagulation. The blood samples were first subjected to soft spin (472× *g*, 10 min) using a horizontal centrifuge (KS-5000; Kubota, Osaka, Japan). The upper plasma fraction and the buffy coat fraction (the lower plasma fraction facing the red blood cell (RBC) fraction) were then subjected to hard spin (578× *g*, 5 min), which was substantially slower than usual to avoid unexpected platelet aggregation, using an angle-fixed centrifuge (Sigma 1–14; Sigma Laborzentrifugen GmbH, Osterode am Harz, Germany). The resulting platelet pellets were gently suspended in PBS. The platelet suspension including white blood cells (WBCs) and RBCs was used only for examination of the effects of WBCs on quantification. These suspensions were aliquoted into 1.5 mL-sample tubes, and stimulated with 0.0125–0.1% CaCl_2_ for up to 60 min at room temperature (18–22 °C). At the end of incubation, platelets were fixed with ThromboFix Platelet Stabilizer (Beckman-Coulter Life Sciences, Indianapolis, IN, USA) [51] and stored at 4 °C for at least 24 h prior to the quantification of polyP.

### 4.4. Quantification of Platelet polyP Levels

Fixed platelets were centrifuged at 578× *g* for 5 min, and the resulting platelet pellets were gently suspended in Milli-Q water. Blood cell and hemoglobin (HGB) levels were determined using an automated hematology analyzer (pocH-100i V Diff; Sysmex, Kobe, Japan). If the platelet counts were higher, the samples were diluted with Milli-Q water to adjust platelet counts within the range of polyP quantification. As the DAPI concentration increases, the range of polyP quantification can be expanded, however, based on the cost-benefit ratio, we adopted 4 μL/sample (4 μg/mL). Platelet suspensions were incubated with DAPI for 30 min at room temperature (18–22 °C) without perforation and were directly subjected to fluorescence measurements using a fluorometer (FC-1; Tokai Optical Co., Ltd., Okazaki, Japan) with excitation and emission wavelengths of 425 and 525 nm, respectively, according to the red-shift fluorescence of DAPI bound to polyP [29].

Notably, when filamentous fibrin clot formation made of intraplatelet fibrinogen was optically observed after the addition of CaCl_2_, the samples were ultrasonicated for 10 s using a Vibra-cell ultrasonic liquid processor (VCX130PB; Sonics & Materials, Inc., Newtown, CT, USA) before treatment with DAPI. A calibration curve for fluorescence intensity against polyP concentration was drawn using synthetic long-chain polyP (C-60; Bioenex, Hiroshima, Japan), while a calibration curve for platelet polyP level against platelet count was based on representative data from a single donor experiment. Similar data were obtained from three independent experiments.

When standard curves using synthetic polyP and ADP were constructed or when the effects of added CaCl_2_ were examined in the absence of platelets, fluorescence quantification was performed in Milli-Q water.

### 4.5. Cytochemical Staining of polyP

PolyP stored in and released from platelets was visualized using DAPI [17,52]. Briefly, the control and activated platelets were immobilized on glass slides using a Cytospin 4 cytocentrifuge (Thermo Fisher Scientific, Waltham, MA, USA) and fixed with 10% neutral buffered formalin for 24 h. After washing with PBS, the platelets were stained with PBS containing 0.1% Tween-20 (FUJIFILM Wako Pure Chemical, Osaka, Japan) and 10 μg/mL DAPI in the presence of Phalloidin-iFluor 555 (1:200 dilution) (Abcam, Cambridge, UK) for 30 min. The specimens were then washed and mounted using an antifade mounting medium (Vectashield; Vector Laboratories, Burlingame, CA, USA) and subjected to microscopic examination using a fluorescence microscope (Eclipse 80i; Nikon, Tokyo, Japan) equipped with a BV-2A filter cube (excitation filter: 400–440 nm; dichroic mirror: 455 nm; barrier filter: 470 nm) to detect DAPI and phalloidin, respectively. 

### 4.6. Statistical Analysis

Each quantification was performed in triplicate. Unless otherwise stated, data are expressed as the mean ± standard deviation (SD) (Figure 2b and Figure 5a–c), scatter plots (Figure 2c and Figure 4a,b) or box plots (Figure 3a and Figure 4c). For the comparison of the two groups, after the data were tested for normality (Shapiro-Wilk) and equal variance (Brown-Forsythe), a parametric two-tailed Student’s t-test was performed (SigmaPlot 13.0; Systat Software, Inc., San Jose, CA, USA). For the comparison of multiple groups, when parametric tests were suggested by both normality and equal variance testing, a one-way ANOVA followed by Bonferroni’s test was performed. Differences were considered statistically significant at *p* < 0.05.

Linear regression analysis and the calculation of correlation coefficient values I were performed using SigmaPlot. Absolute R values ranging from 0.4 to 0.6 and from 0.6 to 0.8 were considered “moderate” and “strong” correlation, respectively. R values of 0.2 or lower indicated a “very weak” correlation.

## 5. Conclusions

In combination with the previous visualization techniques, the simple quantification technique presented in this study represents a powerful tool with which to gain a better understanding of platelet polyP and further develop novel aspects of both the platelet physiology and clinical applications of polyP. 

When applied in the Japanese population, this simple quantitation technique revealed that polyP levels per platelet count are significantly higher in women than that in men and that responsiveness to CaCl_2_ stimulation is also significantly higher in women than that in men. However, some limitations existed in this study, especially in terms of data concerning situations where the female hormone levels can be changed, for example, pregnancy or any disease conditions, in breast cancer patients. Thus, further studies with a larger sample size, including such female subjects, are required to reach a more robust conclusion regarding gender differences. 

## Figures and Tables

**Figure 1 ijms-22-07257-f001:**
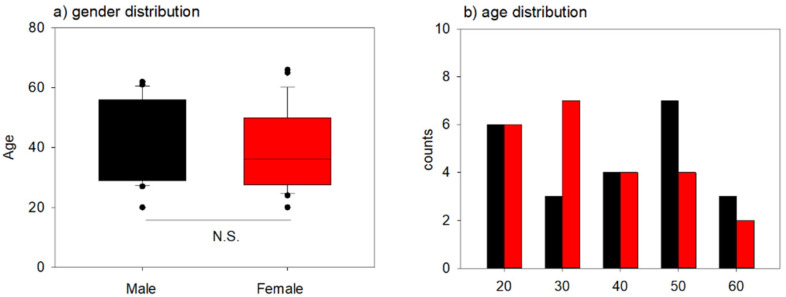
Age distribution in male and female groups. (**a**) Age distribution of male and female groups (*n* = 23 for each). (**b**) Age-dependent histograms of donor counts of both gender groups; “20” on the X-axis represents ages between “20–29”. Similarly, the other labels represent the corresponding age groups. The black and red columns represent male and female groups, respectively. The block dots represent outliers.

**Figure 2 ijms-22-07257-f002:**
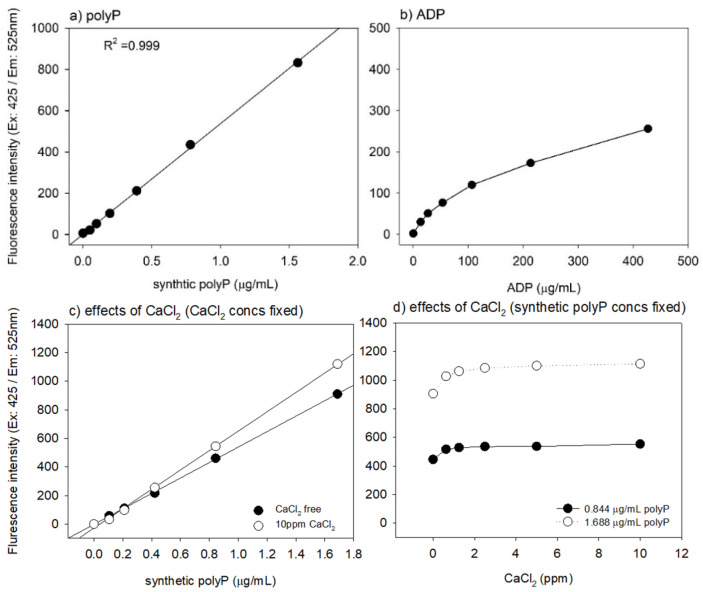
Standard curves for DAPI-reactive fluorescence intensity against synthetic polyP and ADP. (**a**) PolyP: Linear regression is observed in a wide range of polyP concentration. (**b**) ADP: Fluorescence signals are relatively lower even at higher concentrations; the regression is not expressed by a line. (**c**,**d**) Effects of exogenously added CaCl_2_ on polyP quantification. Under the conditions of fixed concentrations of CaCl_2_ (**c**) and polyP (**d**), DAPI-reactive fluorescence intensity against synthetic polyP was quantified.

**Figure 3 ijms-22-07257-f003:**
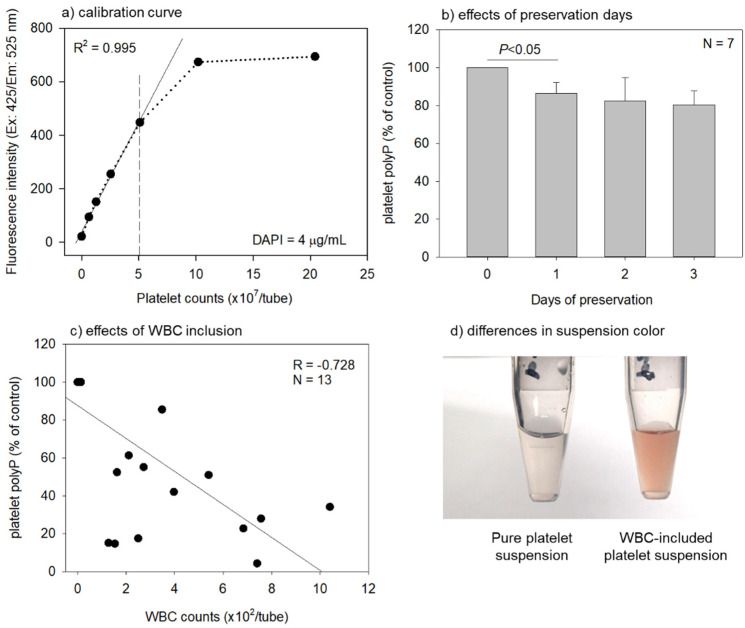
Standard curve for DAPI-reactive fluorescence intensity against platelets and the effects of various factors on polyP quantification. (**a**) A standard curve for the fluorescence signals in platelets probed with 4 μg/mL DAPI. A good linear regression (R^2^ = 0.995) was obtained in the range of platelet counts (up to 5 × 10^7^/tube). (**b**) The effects of preservation days on quantification of platelet polyP levels (*n* = 7). A significant difference was observed only in the first 1-day interval. (**c**) Effects of WBC inclusion on quantification of platelet polyP levels. Since WBCs may shift the original polyP quantification upward, this downward shift may be due to RBC inclusion; a strong correlation (R = −0.728) was observed; *n* = 13. (**d**) The representative difference in suspension color. The suspension including WBC was faintly stained red (right).

**Figure 4 ijms-22-07257-f004:**
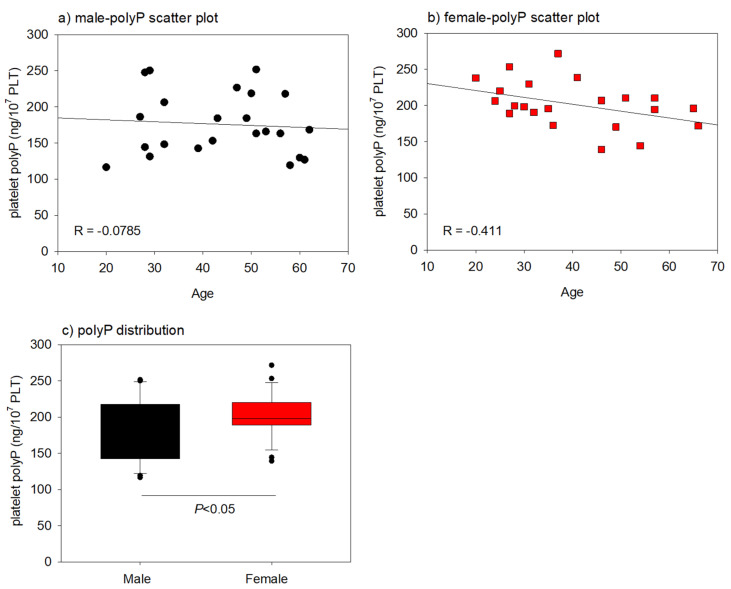
Scatter plots for age vs. platelet polyP levels in male and female groups. (**a**) A scatter plot of male group; a very weak correlation (R = −0.0785) was observed. (**b**) A scatter plot for female group; a moderate correlation (R = −0.411) was observed. (**c**) Comparison of platelet polyP levels between male and female groups; a significant difference was obtained (*n* = 23 for each).

**Figure 5 ijms-22-07257-f005:**
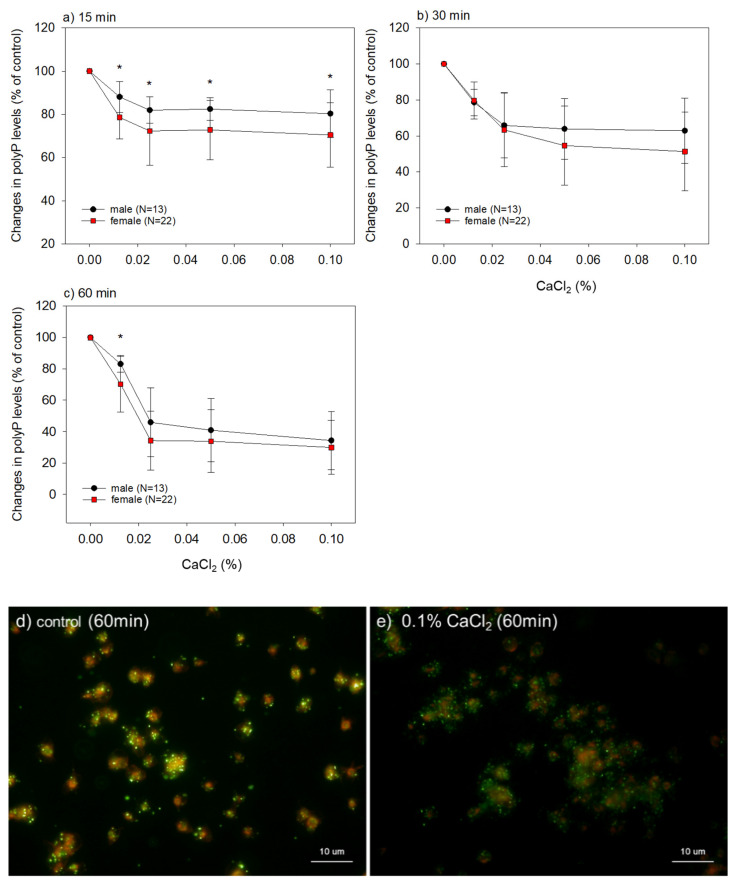
(**a**–**c**) Concentration-response effects of CaCl_2_ on platelet polyP levels in the male and female groups. The platelet polyP levels were quantified at (**a**) 15, (**b**) 30, and (**c**) 60 min after 0.0125–0.1% CaCl_2_ stimulation (*n* = 13 (male group) or 22 (female group)). The asterisks represent statistically significant difference between male and female groups at the same CaCl2 concentrations (*p* < 0.05). (**d**,**e**) Cytochemical visualization of polyP particles in platelets derived from a male donor. Platelets were stimulated with 0.1% CaCl_2_ for 60 min (**e**) or Milli-Q water (10% volume) as control (**d**). The polyP and polymerized actin were stained green and red, respectively. Similar observations were obtained from other three independent experiments using samples from both genders.

## Data Availability

The datasets used and/or analyzed during the current study are available from the corresponding author on reasonable request.

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
