# Peer review of "Fluorometric Quantification of Human Platelet Polyphosphate Using 4′,6-Diamidine-2-phenylindole Dihydrochloride: Applications in the Japanese Population"

_ijms, 2021, doi:10.3390/ijms22147257_

Round 1

Reviewer 1 Report

In their study, Watanabe et al, using a well-known assay, called DAPI probe uptake in platelets, measure polyphosphates. The novelty in this study is that the authors performed these assays in platelets from the Japanese population, obtained from both genders female and male. The authors found that polyphosphate levels are significantly high in women than in men. In addition, they found a negative correlation between age/polyphosphate levels. This is a simple study, I don’t think that it brings something very new, a part that polyphosphate levels can be dependent on age and gender in Japan. This study could be relevant to perform in the situations where the hormone levels can be changed, for example, pregnancy or any disease conditions, in breast cancer patients. 

Author Response

- This is a simple study, I don’t think that it brings something very new, a part that polyphosphate levels can be dependent on age and gender in Japan. This study could be relevant to perform in the situations where the hormone levels can be changed, for example, pregnancy or any disease conditions, in breast cancer patients.

Response: Thank you for your valuable comments. As per your comment, this was a simple study. However, with respect to the optimization of the quantification protocol, we believe that the study has merit for publication. Based on previous reports (Omelon et al. Biochem. Soc. Trans. 2016;44:46-9, among others), we selected the wavelengths for analysis. However, we spent several months in optimizing the experimental conditions and eventually found the ideal conditions regarding the fixative, fixation time, platelet washing, solution, and probing time.

To date, purification using specific columns is thought to be necessary for the exclusion of interfering factors and the accurate quantification of cellular polyP levels. However, an increase in the number of processing steps will generally increase the amount of technical biases and decrease the recovery of target molecules and data reliability. Therefore, if fluorescence microplate readers are used, we believe that this protocol will be useful for high-throughput screening.

In our early experimental planning, we aimed to show the applicability of our optimized protocol using “predictable” data. To our knowledge, the quantification of platelet polyP levels has been rarely reported or is not yet established as clinical data in Japanese populations compared with populations in other countries. However, if polyP levels depend on mitochondrial activity for ATP generation as indicated elsewhere, it could be hypothesized that platelet polyP levels are higher in younger females than in older females or males.

As expected, we successfully demonstrated that platelet polyP levels can be quantified by optimizing the protocol without purification processes and applying this protocol to collect clinical data.

Regarding the reviewer’s suggestions on examining the possible correlations between hormone levels and disease conditions, we plan to conduct future studies on this topic. However, to investigate such matters, we decided to first establish the fundamental concepts and study the corresponding processes step-by-step. Thus, this study is the first step toward a more in-depth investigation to clarify the causes of this phenomenon.

Reviewer 2 Report

Watanabe et al., optimized a polyp quantification method using DAPI to measure polyp levels in platelets of a Japanese population. This study shows that the polyP levels were higher in platelets of female donors compared to those of male donors.

Major concerns

  • The findings of this paper provide minimally incremental advances to the field and does not sufficiently differ their previous publication by Sato et al., 2021, IJMS. The differences between the two techniques described in methods “cytochemical visualization” and subsequent quantification and “quantification of polyP levels” are unclear.
  • Why is fibrin clot formation being detected? Activating washed platelets with calcium should not result in fibrin formation, as the coagulation factors should be washed away from the samples. What is the source of fibrinogen and thrombin?
  • The authors mention that polyP levels may be marker of platelet concentrate quality and state this as a potential goal of the study. Does it correlate with PC quality? If so, how does it compare to other markers of PC quality?
  • Identifying the absolute polyP levels in platelets, instead of relative to control, by comparing against a standard curve developed using known levels of polyP (length to those found in platelets) would provide much informative dataset for future analysis.
  • Does adding calcium to purified polyP change the fluoresecent readings when polyP binds to DAPI?

Minor concerns

  • Figure 3a: label should be “calibration curve”.
  • Figure 5e: the “2” in CaCl­2 should be subscripted.
  • Figure 5 legends: specify the what each color represents in figures d and e.

Author Response

Major concerns

- The findings of this paper provide minimally incremental advances to the field and does not sufficiently differ their previous publication by Sato et al., 2021, IJMS. The differences between the two techniques described in methods “cytochemical visualization” and subsequent quantification and “quantification of polyP levels” are unclear.

Response: In a previous study (Sato et al. IJMS, 2020), we optimized the cytochemical protocol for polyP visualization and demonstrated the distribution of polyP in both control and activated platelets. This previous study demonstrated polyP release by activated platelets in a qualitative manner, whereas the present study quantified polyP levels in and on platelets. We believe that there is a major difference between the findings of the previous and present studies.

With respect to the methodological tip, we used a phosphate-containing buffer for cytochemical visualization to efficiently retain polyP in platelets. However, in the present study, because the presence of phosphate increases the basal levels of polyP in the samples, we selected Milli-Q water. Although we did not emphasize this difference in the text, we believe that this may be very helpful information for non-experts in the field of polyP research.

- Why is fibrin clot formation being detected? Activating washed platelets with calcium should not result in fibrin formation, as the coagulation factors should be washed away from the samples. What is the source of fibrinogen and thrombin?

Response: It is difficult to optically detect platelet aggregation without using microscopy. However, when fibrin is formed and becomes attached to the aggregated platelets, white filamentous substances can be observed macroscopically with the naked eye.

A major source of fibrinogen is plasma; however, some fibrinogen is taken up by circulating platelets and stored in their alpha granules (Blood 2011;118(5):1190–91).

Thrombin, a form of prothrombin, circulates in the blood and is not stored in platelets. However, contamination of prothrombin or thrombin in the washed platelet suspension could not be completely excluded because extensive washing was avoided to save polyP on the platelet surface. In addition, thrombospondin, the major alpha granule protein of human platelets, serves to stabilize fibrinogen binding to the activated platelet surface (J Clin Invest 1984;74(5):1764–72).

Because prothrombin is converted to thrombin on the activated platelet surface (Blood 2011;117(5):1710–18), it is conceivable that a trace amount of prothrombin in the platelet suspension (compared with that in plasma) could react with fibrinogen that is trapped and stabilized by thrombospondin on the platelet surface to generate fibrin fibers and consequently filamentous fibrin clots.

- The authors mention that polyP levels may be marker of platelet concentrate quality and state this as a potential goal of the study. Does it correlate with PC quality? If so, how does it compare to other markers of PC quality?

Response: PC is a cocktail containing many bioactive factors, some of which function positively for tissue regeneration, attenuate tissue regenerative action, or interfere with the positive factors. Thus, although high-quality PC can simply be defined as having a higher potential for tissue regeneration, it is not easy to biochemically define PC quality in detail.

Some researchers have proposed that blood cell counting can be used to characterize individual PCs. However, these counts are one of their many characteristics. Several other characteristics remain to be evaluated, and polyP is just one of these characteristics of PCs at present. Our study on PC quality is still in its infancy. We have been accumulating data step-by-step to narrow down the broad characteristics of PC and propose the major criteria of determining PC quality in the near future.

- Identifying the absolute polyP levels in platelets, instead of relative to control, by comparing against a standard curve developed using known levels of polyP (length to those found in platelets) would provide much informative dataset for future analysis.

Does adding calcium to purified polyP change the fluorescent readings when polyP binds to DAPI?

Response: We have performed this additional experiment to examine the effects of CaCl2 on polyP quantification. Inclusion of CaCl2 at a concentration of 10 ppm amplified DAPI signals by approximately 15%–25% in the range of platelet polyP levels. At present, there is no evidence to evaluate the significance of this quantification artifact for appropriate data interpretation. However, this is the possibility in the scenario worst. We performed careful manipulation to maximally exclude Ca and Pi from the assay solution.

This finding has been added to Figure 2

Minor concerns

- Figure 3a: label should be “calibration curve”.

Response: Thank you for this comment and the following comments. We have corrected this label accordingly.

- Figure 5e: the “2” in CaCl­2 should be subscripted.

Response: We have corrected it.

- Figure 5 legends: specify the what each color represents in figures d and e.

Response: We have added the explanation.

Reviewer 3 Report

In this study, the well-known fluorescent dye DAPI was used to quantify platelet polyphosphate (polyP) in platelets. Although DAPI is better known as a stain for DNA, it can also form complexes with enhanced and red-shifted fluorescence in the presence of other polyanions such as polyP (and also heparin), with emission at about 550 nm rather than 460 for DNA.  

The authors have developed and verified a simple protocol for quantitative determination of polyP in platelets and have investigated the effects of sample storage and of the presence of other blood cells. Platelet polyP was then measured for a small group of male and female subjects. Female subjects had significantly higher values than male subjects for platelet polyp, and there wa a negative correlation with age, though not strong.

The DAPI method is here used appropriately, with due consideration for interfering substances. It is surprising that the authors do not emphasise the importance of avoiding the use of heparin as anticoagulant. It can also be recommended that the authors cite the useful short review of Omelon et al. 2016 (Biochem. Soc. Trans. 44:46-9). One typographical error, in line 128: should the word ‘lay’ be ‘light’?

Author Response

- The DAPI method is here used appropriately, with due consideration for interfering substances. It is surprising that the authors do not emphasis the importance of avoiding the use of heparin as anticoagulant. It can also be recommended that the authors cite the useful short review of Omelon et al. 2016 (Biochem. Soc. Trans. 44:46-9).

Response: From a biomedical point of view, heparin is not recommended for the preparation of PRP for regenerative therapy applications. Heparin is a potent anticoagulant used to treat patients with thrombosis and to prevent unexpected coagulation during dialysis. Thus, heparin is mainly used in vivo.

In contrast, for in vitro use, EDTA, ACD, or NaF have been used to prevent coagulation in collected blood samples for laboratory testing. Especially for PRP preparation, ACD has been recommended and is widely used in clinical settings (Harrison, J Thromb Haemost 2018;16:1895-1900). In our previous study, we aimed to re-evaluate these anticoagulants in PRP preparation (Aizawa et al., Biomedicines. 2020;8:42) and found that heparin induces platelet aggregation. This finding led us to conclude that heparin is not suitable for PRP preparation; thus, we have never used heparin for this purpose. Accordingly, we emphasize the importance of avoiding heparin use. In the revised version, however, we have added this important caution to Omelon’s paper (BST, 2016) in the Discussion section.

Thank you for your valuable comment. This seems to be a special issue for polyP that has never been identified. We have cited this article in the Materials and Methods section.

- One typographical error, in line 128: should the word ‘lay’ be ‘light’?

Response: Thank you for this advice. We have corrected it.

Round 2

Reviewer 2 Report

Thank you for addressing the previous concerns 

Author Response

Thank you very much for your valuable comments and evaluation.